# Over-expression of *TaDWF4* increases wheat productivity under low and sufficient nitrogen through enhanced carbon assimilation

Matthew J. Milner [1] [✉], Stéphanie M. Swarbreck[1,2], Melanie Craze[1], Sarah Bowden [1], Howard Griffiths[2], Alison R. Bentley [1,3] & Emma J. Wallington[1]

There is a strong pressure to reduce nitrogen (N) fertilizer inputs while maintaining or increasing current cereal crop yields. We show that overexpression of *TaDWF4-B*, the dominant shoot expressed homoeologue of *OsDWF4*, in wheat can increase plant productivity by up to 105% under a range of N levels on marginal soils, resulting in increased N use efficiency (NUE). We show that a two to four-fold increase in *TaDWF4* transcript levels enhances the responsiveness of genes regulated by N. The productivity increases seen were primarily due to the maintenance of photosystem II operating efficiency and carbon assimilation in plants when grown under limiting N conditions and not an overall increase in photosynthesis capacity. The increased biomass production and yield per plant in *TaDWF4* OE lines could be linked to modified carbon partitioning and changes in expression pattern of the growth regulator Target Of Rapamycin, offering a route towards breeding for sustained yield and lower N inputs.

[1] NIAB, 93 Lawrence Weaver Road, Cambridge CB3 0LE, UK. [2] Department of Plant Sciences, University of Cambridge, Cambridge CB2 3EA, UK. [3]Present address: International Maize and Wheat Improvement Center (CIMMYT), Texcoco, Mexico. [✉]email: matthew.milner@niab.com

Wheat (*Triticum aestivum* L.) is a major crop worldwide providing 23% of human dietary protein[1]. In many parts of the world, on-farm wheat yields have plateaued since the mid-1990s, although the yield capacity of cultivars has continued to increase[2]. Modern agricultural practices include the development of high-yielding varieties of cereal grains, expansion of irrigation infrastructure, modernization of management techniques, distribution of hybridized seeds, and the addition of synthetic fertilizers and pesticides to increase and protect yields. However, this approach has led to direct losses of reactive N to the aquatic environment which lower water quality through eutrophication, coupled with the loss of $N_2O$ and the high $CO_2$ emissions during fertilizer production, so the environmental consequences are substantial[3–5]. As it has been estimated that only 33% of N applied to a field is taken up by the crop, a multi-faceted approach is crucial to reducing the necessary application of fertilizer as well as losses of the applied fertilizer[6,7]. Improving N use efficiency (NUE) (defined as the ratio of grain produced per unit of N supply) can be achieved by improving yield under a constant N supply. Thus far, strategies to improve the yield capacity of crop cultivars have focused on photosynthesis[8–10], with gains in biomass and yield reaching 40%[11,12]. However, this is not sufficient, and reducing our reliance on synthetic fertilizer will also rely on identifying ways to reduce wheat N requirement, i.e., being able to produce wheat with lower N input without a reduction in yield or grain quality[2]. While identifying genes and pathways regulating specific aspects such as N uptake, assimilation or remobilization can provide some advantages, approaches that target regulatory components of N metabolism are more promising. In particular, understanding and identifying genes that can regulate N responsiveness, i.e., the overall capacity of plants to induce morphological and physiological changes according to the external availability of N, are eagerly sought[6,13].

Many domesticated crop species are polyploid and it is thought that the genome duplication events and the resultant chromosome rearrangements have been a major driving force creating biological complexity, novelty, and adaptation to environmental changes[14–16]. Wheat, an allohexaploid with genome AABBDD, is the product of two ancient hybridization events: *Triticum uratu* (A) and *Aegilops spp.* (B) hybridized 0.5 million years ago[17], to form the tetraploid *Triticum turgidum*, which then underwent a second hybridization ~10,000 years ago with *Aegilops tauschii* (D) to form *Triticum aestivum*[18]. A more nuanced process of continuous diversification and allopolyploid speciation to produce the network of polyploid *Triticum* wheats has also been described[19] and with the advent of whole-genome resequencing, the B-subgenome has been shown to exhibit most variation with multiple alien introgressions and deletions[20]. The presence of homeoalleles in hexaploid wheat increases both coding sequence variation per se and regulatory variation with the new promoter and transcription factor combinations and functional diversification of the duplicated genes[21,22]. This genome asymmetry and variation have consequences for quality traits such as the control of wheat seed storage proteins, biotic and abiotic traits, agronomic traits, and response to pests and diseases[23–25].

Phytohormones are known to modify agronomically important traits, in particular, brassinosteroids (BR)[26,27]. BR have been shown to positively affect traits including tiller number, leaf size and angle, photosynthesis, and yield[28–31] plus other desirable traits, including NUE, disease resistance, and end-use quality traits[31–34].

Manipulating levels of *DWF4/CYP90B*, the rate-limiting step in the BR synthesis pathway, has been shown to increase yield, biomass production, tiller number, and quality traits in diverse plant species[30,31,34,35]. The mechanism of how altering BR levels impacts yield is still not completely understood, but in many plant species an increased rate of carbon fixation has been observed, suggesting that photosynthesis is key to the increased yields[30,34].

An explanation of how the BR pathway controls so many different agronomically important phenotypes was shown through the direct interaction between BZR1 (Brassinazole Resistant 1), the transcription factor activating BR related genes, and the Target Of Rapamycin (TOR), the growth regulator sensing the carbon (C) status of the plant[36,37]. Overexpression of TOR in plants also mirrors many of the same phenotypes as modification of the BR pathway, suggesting that TOR may be the mode of increased growth and yields shown by many BR overexpressors. The BR and TOR pathways directly interact through the BR transcription factor BZR1 and the TOR kinase. The TOR kinase can maintain the activation potential of BZR1 and its role in growth promotion, by keeping the BZR1 protein stable[37]. Reducing *TOR* expression by RNAi silencing led to a decreased ability of BR regulated genes to be upregulated, arrested plant growth and, abolished the ability of increased BZR1 to promote growth when high levels of sugar were present[38].

While many studies have shown that modification of the BR pathway can increase yields, plants have typically been assessed under nutritionally replete conditions. In order to assess if the same yield gains could be achieved in wheat grown on either marginal soils, or with lower fertilizer inputs, we generated overexpression lines to test whether modification of *DWF4* levels could drive productivity gains and/or increase NUE.

## Results

**The wheat DWF4 gene family has seven members.** To identify putative orthologs of DWF4/CYP90B in wheat, the amino acid sequence for the rice DWF4/CYP90B (locus Os03g0227700) was used in a BLASTP search using wheat genomics resources at Ensembl (http://plants.ensembl.org/Triticum_aestivum/Info/Index). Seven amino acid sequences were identified with high homology to the rice DWF4 amino acid sequence (e value < $1e^{-50}$, and >70% homology), these were encoded by four sequences located on chromosome 3 (one on chromosome 3A, one on chromosome 3B, two sequential copies on chromosome 3D) and three sequences located on chromosome 4 (Suppl. Fig. 1a). The amino acid sequences were clustered by chromosome group (Suppl. Fig. 1a). Putative proteins encoded by sequences on chromosome group 4 tended to be more similar to each other (98.62–99.01% identity) compared to those on chromosome group 3 (87.97–95.77% identity). Within chromosome group 3, TaDWF4-3D2 was the most distinct within the family (Suppl. Table 1).

Comparison of the group 4 homoeologue putative protein sequences revealed that 4A contains two large insertions when compared to the other homoeologues and OsDWF4 (Suppl. Fig. 2). The first insertion is located between exons 1 and 2, and the second insertion is located between exons 5 and 6, relative to the predicted CDS in the B and D homoeologues. The genes on chromosome 4 showed the highest expression levels, in particular *TaDWF4-4B*, while low to no expression could be seen for any of the genes on chromosome 3 in the wheat expression.com databases[39] (Suppl. Fig. 1a). The expression patterns of the identified homoeologues on chromosome 4 showed unbalanced expression with *TaDWF4-4B* dominating expression in the shoot (70.9%), and *TaDWF4-4A* dominating expression in the spike (56.6%). There was no dominant homeologue expressed in the roots as *TaDWF4-4A* and *TaDWF4-4B* showed similarly but the higher expression in the roots than *TaDWF4-4D*.

**DWF4 gene duplications in wheat.** In diploid plant species where the DWF4 gene family has been characterized (including

Arabidopsis, rice, and maize (*Zea mays*), DWF4 is encoded by a single gene[28,40,41]. It is interesting that in hexaploid wheat, cv. Chinese Spring, seven putative orthologs could be identified, suggesting that there may have been one or possibly two duplication events: the first duplication event leading to the presence of DWF4 genes on both chromosome 3 and chromosome 4 groups, and the second duplication event leading to two DWF4 genes on chromosome 3D, as the three gene models are co-linear on chromosome 3D.

The *DWF4* inter-chromosomal duplication event in wheat, which led to the presence of the DWF4 gene on both chromosome groups 3 and 4, cannot be found in crop or model species such as Arabidopsis, rice, maize, or brachypodium (*Brachypodium distachyon*) (Suppl. Fig. 1b). In rice, the closest related proteins to OsDWF4 are other cytochrome P450s more commonly known as CPD1 or CPD2 sharing 46.7 percent identity at the amino acid level (Ensembl v 46). In Arabidopsis, the closest protein to AtDWF4 (AT3G50660) is also a cytochrome P450, CYP72A1 sharing 36.1% identity at the amino acid level (Ensembl v 46). In maize and brachypodium the closest related proteins are ZmCYP724A1 (Zm00001d003349) with 50.5% identity and BdCYP724A1 (BRADI_5g12990v3) with 47.4% identity.

The inter-chromosomal duplication of the DWF4 genes can also be inferred for both *T. urartu* and *Aegilops tauschii*, two of the three wheat progenitor species (progenitors of the A and D genomes in *T. aestivum*), as well as in the tetraploid *T. dicoccoides* (Suppl. Figure 1b). An intra-chromosomal duplication in *Aegilops tauschii* on chromosome 3, supports the hypothesis that these copies arose before the progenitors hybridized. Therefore, one gene duplication event seems to have arisen before the progenitor species hybridized, and a second event after the hybridization occurred to produce modern bread wheat.

**Overexpression of TaDWF4-4B in wheat leads to the upregulation of BR responsive genes.** Given that *TaDWF4-4B* in Chinese spring showed the highest expression levels in the shoot tissues identified in public RNAseq databases, we selected this homeologue for overexpression in hexaploid wheat (Suppl. Fig. 1). A total of 40 independent $T_0$ plants were generated in the cv. Fielder, with *TaDWF4-4B* CDS, expressed from the constitutive promoter, OsActin[42]. In six out of nine $T_0$ plants with a single T-DNA insertion, *TaDWF4* transcript levels (including both transgene and endogenous gene) were significantly increased from 1.6 to 4.9-fold compared to wild type (Suppl. Fig. 3a). In the four over-expressing (OE) lines showing the highest *TaDWF4* transcript levels, we measured the transcript levels for EXORDIUM (EXO), DWF3/CPD/CYP90A, BZR2, and BES1, as these genes either have been shown in Arabidopsis to be upregulated by BR[43] or involved in the BR pathway[44–46]. *TaEXO* and *TaBES1* transcript levels were significantly higher than WT in the four OE lines tested ($p < 0.05$) and followed the same trend as those of *TaDWF4* (Suppl. Fig. 3b). *TaBZR2* and *TaDWF3* did not show significantly increased expression in OE wheat lines and this contrasts with previous reports in other plant species of either *DWF4* overexpression or the application of exogenous BL[44,47].

**Overexpression of TaDWF4-4B increases productivity under low to high N levels.** The four highest expressing transgenic lines (OE-1–OE-4) and a corresponding null segregant (WT) were grown in pots on a low fertility soil supplemented with $NH_4NO_3$ to reach field-equivalent N levels of 70, 140, and 210 kg/ha N. A low fertility soil with limited N was used as a substrate in order to demonstrate N deficiency symptoms. The lowest level (70 kg/ha N) corresponds to an N deficiency and the highest level (210 kg/ha N) to the agronomic level typical of current UK agronomic practice[48]. All four OE lines

tested showed significant increases in yield per plant when compared to the WT at all 3 N levels ($p = 0$; Fig. 1a). There was also a large increase in above-ground biomass ($p = 0$), ranging from a 45 to 101% increase in the OE lines compared to WT (Fig. 1b). An overall increase in tiller number ($p = 0$; Fig. 1c), grain number per plant ($p = 0$; Fig. 1e) and grain weight measured as thousand grain weight (TGW) ($p = 0$; Fig. 1f) were also observed. The OE lines also showed a significant increase in harvest index (HI, defined as the proportion of biomass in the grain divided by the total biomass) relative to WT ($p = 0$; Fig. 1d). The significant change in yield per plant under all levels of N tested resulted in a significant increase ($p = 0$) in the NUE in OE lines relative to WT (Suppl. Fig. 5), as expected with yield increase. Full statistical comparisons are included in Suppl Tables 2–7.

As anticipated, increasing N supply led to increased yield in both WT and OE lines. The WT line showed a defined N-dependent yield increase, with OE lines producing higher yield at all N levels and generally mirroring the gains seen by WT. The converse effect was seen for biomass where increased N levels decreased the differences seen in biomass; as N became less limiting the differences in biomass decreased (Fig. 1b). Overexpression of *TaDWF4-B* in cv. Fielder led to significantly increased growth compared to the WT even under low N conditions and also led to higher productivity per plant.

**Nitrogen responsive genes in TaDWF4 OE lines show altered expression.** At the whole plant level, OE lines appear to be highly responsive to low N levels, maintaining yield increases under N-deficient to replete conditions. The expression of four genes known to be differentially regulated under high or low N conditions in wheat was selected to test the N-responsiveness at the transcript level of an OE line (OE-1) relative to WT[49,50]. The genes selected encode the high and low-affinity N uptake transporters (NRT2.1 and NRT1), an N transporter involved in N translocation through the plant (NPF7.1), and glutamate dehydrogenase (GHD2), an enzyme involved in N remobilization in response to limiting carbon (C). All four genes tested show a clear increase in OE-1 and WT transcript levels under low N compared to replete N conditions in both shoots and roots of 10-day-old seedlings grown in a hydroponic system (Fig. 2).

Transcript abundance in OE-1 shoot tissues under N limited conditions showed significantly lower expression of *TaNRT1* ($p = 0$), *TaNPF7* ($p < 0.001$) and *TaGDH2* ($p < 0.001$), but not *TaNRT2.1* ($p = 0.07$), compared with WT. Under replete conditions, no significant differences were found between OE-1 and WT. A similar pattern was seen in root tissues under N limited conditions with significantly lower expression of *TaNRT1* ($p = 0$), *TaNPF7* ($p < 0.001$), *TaNRT2.1* ($p < 0.001$), and *TaGDH2* ($p = 0$), in the OE-1 line compared with WT. Under replete conditions only *TaNRT1* and *TaNRT2.1* showed significantly lower expression in OE-1 compared with WT ($p < 0.01$, $p < 0.05$).

Given the putative role of DWF4 in N response we also measured the level of transcripts for *TaDWF4* from chromosome 4 in the shoots and roots of WT Fielder plants grown under four N levels (0, 1/3, 2/3, and full as a proxy for 0, 70, 140, and 210 kg/ha fertilizer application rate). Transcript levels of *TaDWF4* were higher in the shoots relative to the roots of wheat plants grown under the four N levels ($p < 0.05$, Suppl. Fig. 6). Increased N concentrations led to higher *TaDWF4* transcript levels in the shoots and lower transcript levels in the roots (Suppl. Fig. 6, $p < 0.05$). The effect of N level on *TaDWF4* transcript levels was driven by the difference between the lowest N levels. There were no significant differences in transcript levels observed amongst the 1/3N, 2/3N, and N full treatments. *TaDWF4* expression, therefore, appears to be responsive to changes in N levels under

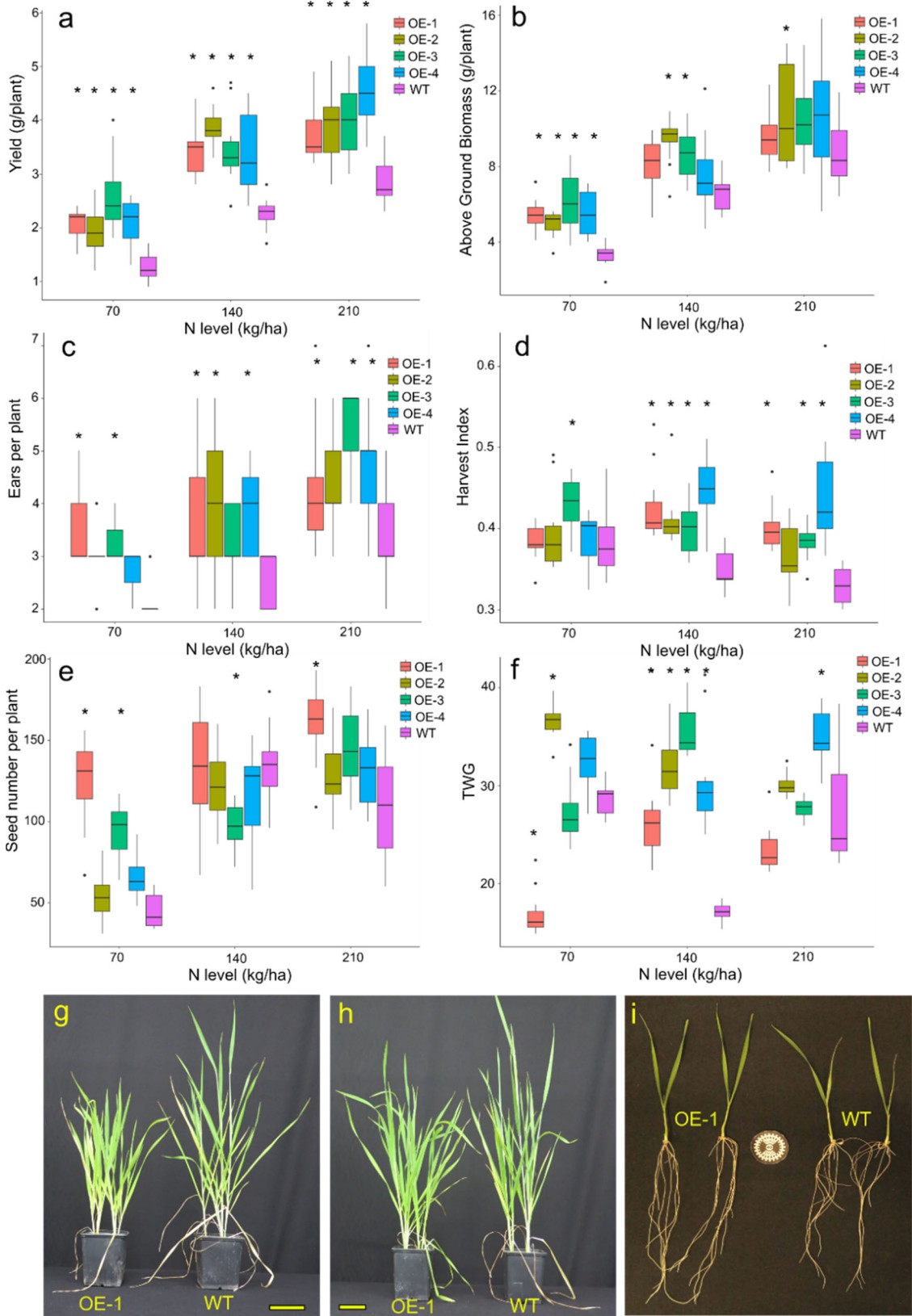

very limiting N conditions promoting expression in the roots and decreasing expression in the shoots in WT Fielder.

**Increases in yield and biomass in TaDWF4 OE lines are independent of N uptake and translocation.** To further understand how the OE lines were driving an increase in yields, the flux of N was measured using [15]N tracer studies. Uptake and translocation of N were measured between the four OE lines and the WT when grown under low N conditions for 2 weeks and supplied with[15]N to quantify short-term uptake and translocation from the root to the shoot. No significant differences were seen in

**Fig. 1 TaDWF4-B overexpressing lines show higher yield, above-ground biomass, seed number per plant, and TGW and harvest index when grown under three N levels (eq. 70, 140, 210, kg/ha).** Data are shown as mean values (central line), lower and upper quartiles (box), minimum and maximum values (whiskers), and outliers as individual points. Fifteen plants were grown per replicate for each genotype by N level combination. The overall plant growth experiment was replicated twice; data from one replicate is shown. The statistical analysis was performed with ANOVA and post hoc Tukey test, asterisk denotes *p* val < 0.05. **a** yield (g) per plant; **b** above-ground biomass (g) per plant; **c** ears per plant; **d** harvest index; **e** seed number per plant; **f** thousand-grain weight per plant (TGW); **g** OE-1 (left) and WT (right) plants grown on 70 kg/ha equivalent; **h** OE-1 (left) and WT (right) plants grown on 210 kg/ha equivalent; **i** 10-day-old plants from OE-1 (left) and WT (right) grown in hydroponic solution under at 70 kg/ha N. Yellow bars in **g** and **h** = 10 cm, size standard in **i** = 40 mm diameter. Full statistical comparisons are included in Supplementary Data 2.

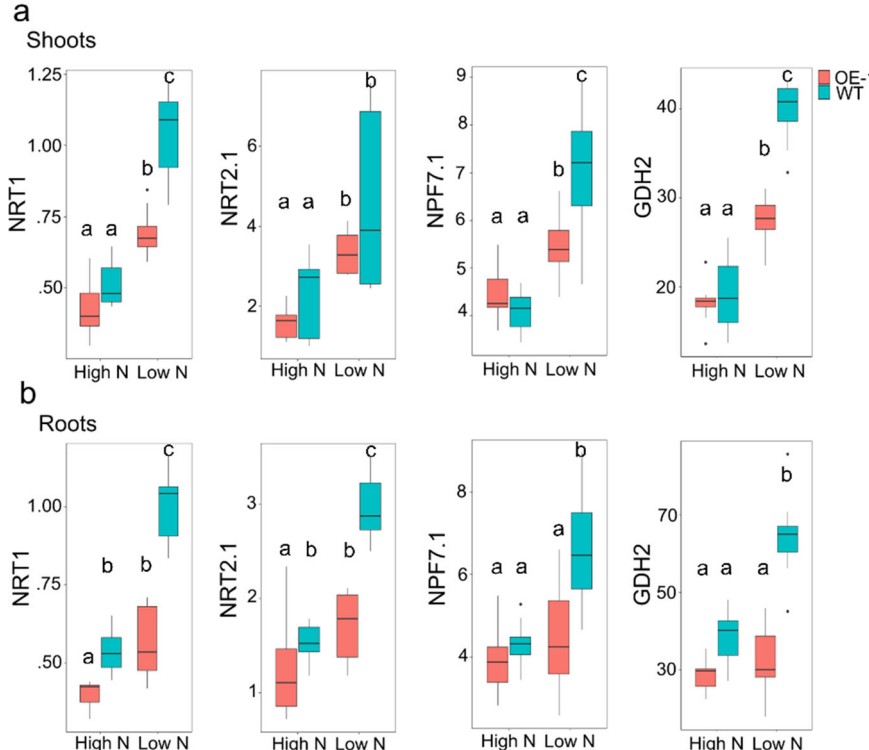

**Fig. 2 Transcript abundance for N regulated genes in shoots and roots of 10-day-old OE-1 and WT seedlings under N limiting and N replete conditions.** Transcript levels for NRT1, NRT2.1, NPF7.1, and GDH2 genes in wheat are shown relative to the expression of *TaUbi* under low (LN) and high nitrogen (HN) in hydroponic solution in **a** shoot; and **b** root tissues. Data are shown as mean values (central line), lower and upper quartiles (box), minimum and maximum values (whiskers), and outliers as individual points. The statistical analysis was performed with ANOVA and post hoc Tukey test, letters correspond to significant differences between transcript levels of either line under either treatment (*p* < 0.05).

the uptake or translocation of N between any of the lines tested (Fig. 3a, b) (ANOVA, *n.s.*).

To understand if the OE lines showed altered N content compared with WT over a longer plant growth cycle, total N was measured in the grain and senesced flag leaf from plants grown under three N levels to compare N taken up from the soil at the end of the life cycle. At harvest, no significant differences in flag leaf N content could be measured in plants grown on eq. 210 kg/ha N (Fig. 3c), while the grain N content was significantly lower in the OE compared to WT. The decrease in grain N content in OE lines compared to WT was also measured when plants were grown under three different N levels (Fig. 3d) (ANOVA *p* < 0.05).

**TaDWF4-B overexpression maintains photosynthesis under N limitation.** In rice, OE of *DWF4* led to an increase in leaf-level photosynthetic C fixation[30]. Leaf level $CO_2$ assimilation per unit area was measured on the fourth expanded leaf in the OE-1 and WT lines grown under both low (eq. 70 kg/ha) and high (eq. 210 kg/ha) N levels. No differences in $CO_2$ assimilation could be measured when OE-1 and WT plants were supplied with adequate N and high light (Fig. 4a). A significant increase in $CO_2$

assimilation over a range of $CO_2$ concentrations in the OE-1 line was observed when plants were grown under low N conditions and high light (Fig. 4b). The increase in $CO_2$ assimilation in OE-1 compared to WT under low N was not driven by a significant difference in chlorophyll content, a major sink for N in a plant, as both the OE-1 and WT line had similar levels of chlorophyll under both low and high N (Fig. 4c) (*p* = 0.34). Overall, this suggests that under low N conditions an increased photosynthetic capacity may drive the yield and biomass gains measured in the OE lines. However, this is not the case when plants are grown under N-replete conditions.

To understand if the plants were able to maintain photosynthesis at a higher level when grown under lower input conditions, plants were grown on the same low fertility soil as shown in Fig. 1 supplemented with either 70 or 210 kg/ha equivalent N. Spot measurements of $CO_2$ assimilation were taken under growth chamber conditions, with light levels at 250 µmol m$^{-2}$ s$^{-1}$ and 400 ppm $CO_2$. Under these conditions, a significant increase in $CO_2$ assimilation could be measured in OE-1 compared to WT, under both N levels tested (Fig. 5a) (*p* < 0.05). Measurements of chlorophyll fluorescence were taken as a proxy for the operating

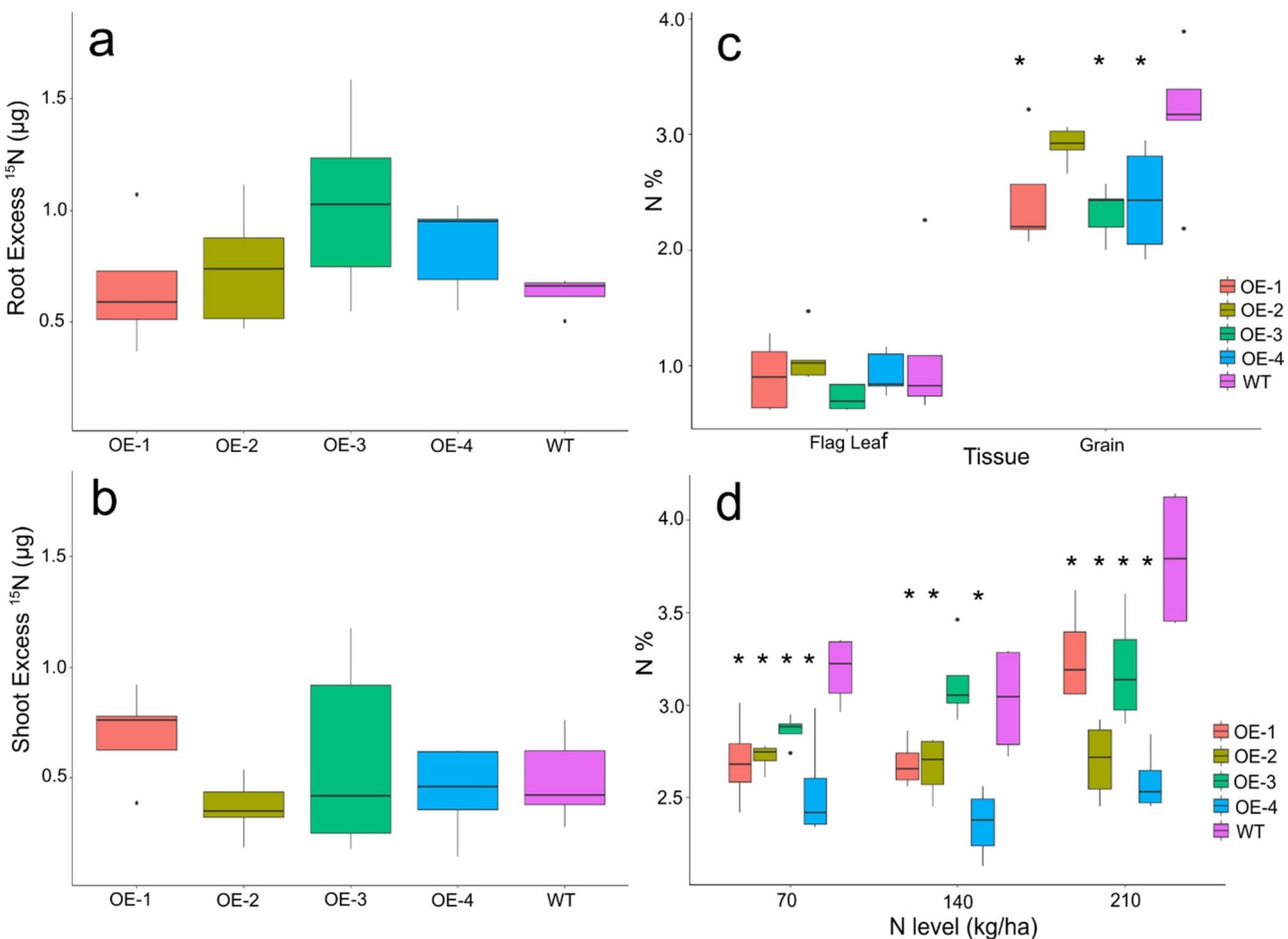

**Fig. 3 Nitrogen flux and concentrations of *TaDWF4-B* overexpression plants. a** Uptake of $^{15}$N in roots of overexpression lines relative to WT roots.
**b** Translocation of $^{15}$N from the soil to the shoot in 5 min of uptake in OE lines relative to WT. **c** Percentage N of flag leaves and grains are grown on
210 kg/ha at harvest. **d** N content of grain grown on three N levels. Data are shown as mean values (central line), lower and upper quartiles (box),
minimum and maximum values (whiskers), and outliers as individual points. The statistical analysis was performed with ANOVA and post hoc Tukey test.
Asterisks indicate a significant difference ($p < 0.05$) between WT and an OE line at the same N level.

efficiency of photosystem II ((Fm'–Fo)/Fm') on light-adapted leaves
from plants grown under low (70 kg/ha) vs. high (210 kg/ha) N
conditions. OE-1 plants had a greater proportion of PSII centers
available to collect the saturating light pulse than that of WT plants
under both low and high N conditions (Fig. 5b) ($p = 0$). While PSII
efficiency declined in WT plants grown under low N conditions
($p < 0.001$), this was not the case for OE-1 plants.

To understand the effect of continued photosynthesis on C
content, flag leaves of fully senesced plants grown on 210 kg/ha
equivalent N were measured. There was a significant increase
(1–2%) in the flag leaf C content in OE-1 lines compared to WT
($p < 0.01$) (Suppl. Fig. 7). Soluble sugars were measured in 14-
day-old seedlings grown in hydroponic solution under low (1/3,
70 kg/ha) or replete (Full, 210 kg/ha) conditions. Under replete N
conditions, WT showed significantly higher levels of both soluble
sugars measured (D-fructose, D-glucose) ($p < 0.05$) (Fig. 6a–d), led
mostly by large differences in the roots of both sugars measured.
Under low N conditions, there were similar levels of bother
sugars per gram dry weight in the shoots of OE-1 plants relative
to WT plants. Although the distribution in the two tissues tested
of the sugars is different and significant differences were observed
in both sugars in both tissues measured ($p < 0.05$).

Next, we tested whether the effect of *TaDWF4-B* over-
expression on photosynthesis and soluble sugar/C levels was
mediated by TOR. Significantly higher transcript levels for

*TaTOR* were measured in the roots of OE-1 relative to WT
when plants were grown under both low and replate N in
hydroponics. *TaTOR* transcripts levels were higher in roots than
shoots in OE-1 ($p < 0.05$) and a significant difference in TOR
expression was seen under both low and high N growth
conditions ($p < 0.05$) (Fig. 6d, e). The effect of low N led to
increased *TaTOR* expression in OE-1 shoots, however, the
converse was seen in WT shoots where *TaTOR* transcripts levels
were markedly reduced under low N. In roots OE-1 showed
significantly higher transcript levels than WT under both high
and low N (Fig. 6e).

**Indirect selection has not increased DWF4 expression in
modern wheat varieties.** To understand if the increased photo-
synthesis observed in the OE lines has already been indirectly
selected for in modern wheat cultivars, seven diverse spring wheat
cultivars, as well as Fielder and OE-1, were tested for both their
ability to maintain PSII operating efficiency under limiting N
levels, and to determine if this could be a factor in determining
NUE differences. The ability of the different varieties to maintain
PSII electron transport could be seen in three different varieties
including the former elite UK spring wheat cultivar Paragon
(Fig. 7a) suggesting that this trait may have been indirectly fixed
via breeding. *TaDWF4* transcript levels were lower in all cultivars
tested compared to OE-1, and there was no significant difference

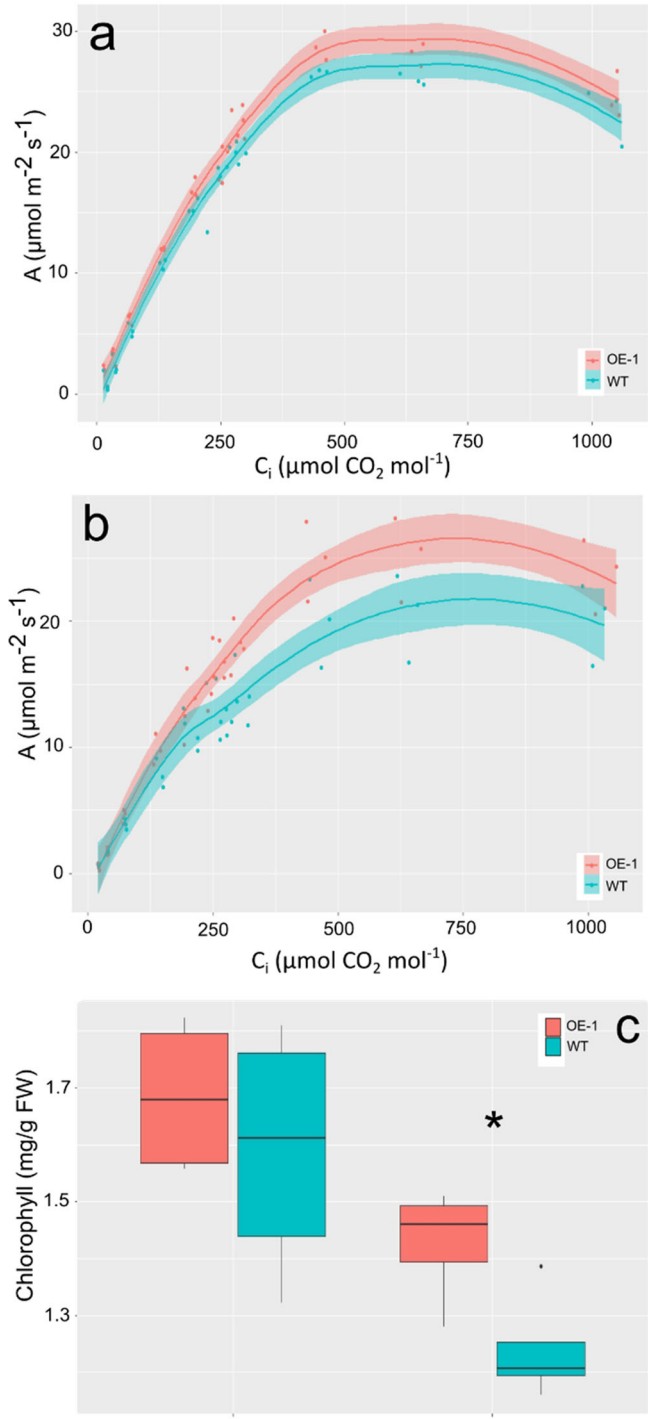

**Fig. 4 Photosynthetic C assimilation in the leaves of *TaDWF4-B* overexpression lines and WT under high (210 kg/ha) and low (70 kg/ha) N levels.** Plants were grown in a growth chamber with a light intensity of 1250 μmol m$^{-2}$ s$^{-1}$ and $CO_2$ level of 400 ppm: **a** A/Ci curve of OE-1 and WT plants grown on under high N conditions (equivalent N 210 kg/ha) or **b** under low N conditions (equivalent N 70 kg/ha). **c** Chlorophyll content in the leaves of plants used for A/Ci measurements. In panels **a** and **b** the shading represents the 95% confidence interval. In panel **c**, data is shown as mean values (central line), lower and upper quartiles (box), minimum and maximum values (whiskers), and outliers as individual points. Data were collected from six individual plants measured on the fourth fully expanded leaf.

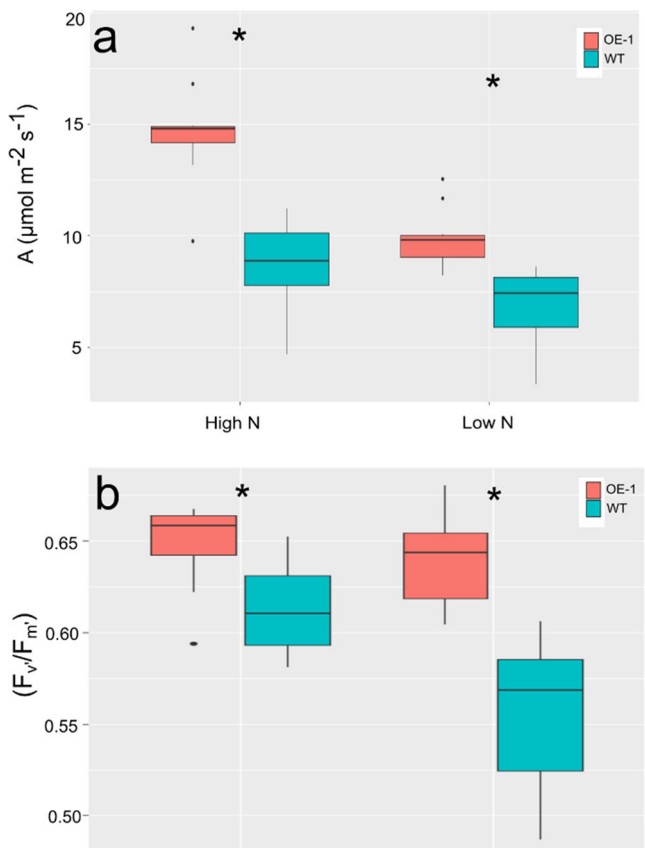

**Fig. 5 Photosynthetic performance of OE-1 or WT plants under low light conditions. a** Spot measurements of C assimilation in OE-1 or WT undergrown in growth chamber conditions which include a light intensity of 250 μmol m$^{-2}$ s$^{-1}$ and CO2 level of 400 ppm. **b** Maximum efficiency of PSII under light (Fv'/Fm') in light-adapted plants grown on two different N levels. Plants were grown on either (Low) 70 or (High) 210 kg/ha N equivalent and the fourth leaf was measured for PSII activity. The statistical analysis was performed with ANOVA and post hoc Tukey test. Data are shown as mean values (central line), lower and upper quartiles (box), minimum and maximum values (whiskers), and outliers as individual points. Data were collected from six individual plants measured on the fourth fully expanded leaf. Asterisk in panels **a** and **b** indicates a significant difference ($p < 0.05$) between WT and OE-1 at the same N level.

amongst cultivars (Fig. 7b). The lack of selection for higher *DWF4* expression is further supported as lines that maintained PSII reaction centers open had both high and low NUE, where NUE is defined as the yield on low N (eq. 70 kg/ha) divided by yield on high N (eq. 210 kg/ha) (Suppl. Fig. 1).

## Discussion

In this study, we identified the functional orthologues of the *OsDWF4* in wheat and generated OE lines to understand whether increased *TaDWF4* expression could be used to increase wheat NUE and drive an increase in yield as seen in other crops with modified *DWF4* expression. We identified seven putative orthologues in wheat, four on chromosome 3 plus three on chromosome 4, and have shown that the putative orthologues on chromosome 3 had low level to no expression, whereas the 4B homoeologue had the highest expression, based on data available from public wheat expression databases. Significant tissue type differences were seen in the expression of the three

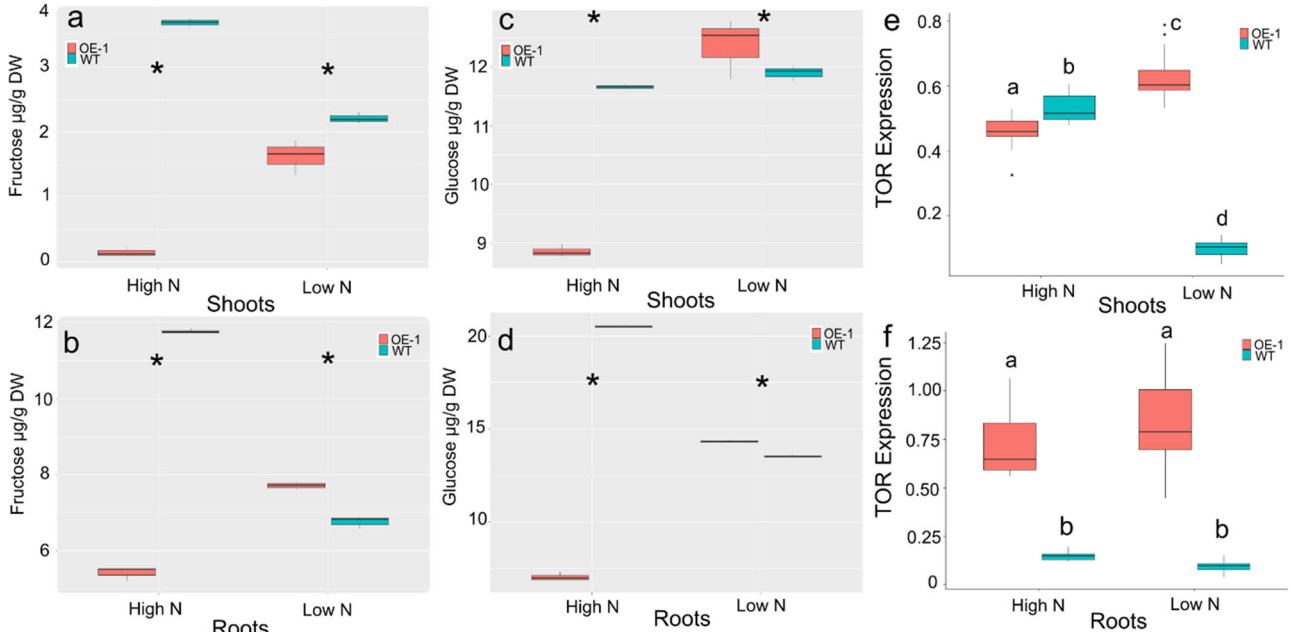

**Fig. 6 Root and shoot glucose and fructose content in OE-1 and WT wheat plants grown under high N or low N conditions.** Plants grown under hydroponic conditions lead to increased transcript levels of *TaTOR* and soluble sugars. Fructose (**a**, **b**), and glucose (**c**, **d**)) were extracted from wheat leaves of 14-day old plants grown High N (Full N) or Low N (0N). The transcript abundance of *TaTOR* was measured in the shoots (**e**) and roots (**f**) in both OE-1 and WT 14-day old wheat plants grown under High N or Low N, the expression shown is relative to *TaUbi*. $n = 3$ plants for each treatment and line testing. The statistical analysis in all panels was performed with ANOVA and post hoc Tukey test. Data in all panels are shown as mean values (central line), lower and upper quartiles (box), minimum and maximum values (whiskers), and outliers as individual points. Asterisk indicates a significant difference ($p < 0.05$) between WT and OE-1 at the same N level. Letters in panels **e** and **f** indicate significant differences ($p < 0.05$) amongst both the line and the treatment.

homoeologues, suggesting that specific regulatory elements may control modification of *TaDWF4* expression in particular tissues to increase plant growth in wheat. The duplication of putative orthologues on the two different chromosome groups also appears to have happened before hexaploid bread wheat arose 5.5 Mya, as each of the progenitor species, durum wheat, and *T. dicoccoides* have the duplication[24]. Further support for the orthologues on chromosome 4 is similar to the previously characterized DWF4 is the high degree of synteny between wheat chromosome group 4, and rice chromosome 3, on which OsDWF4 resides, extending most of its length. In contrast, chromosome group 3 is the best conserved of all wheat chromosome groups, along both short and long arms, compared with rice chromosome 1[51].

Based on the bioinformatic analysis the homoeologue on 4B was selected for the creation of OE lines in order to understand the role of increased *TaDWF4* expression on increasing yield and NUE. Overexpression of *TaDWF4-B* resulted in dramatic increases in several agronomic measures including yield and biomass across a range of N levels. Interestingly, total biomass levels of WT plants started to approach those of OE lines as N levels increased, however, this alone did not lead to a decrease in the difference in yields suggesting that biomass alone is a proxy and not a perfect measure of ultimate yield potential (Fig. 1a, b). The OE lines showed significant increases in yield at all tested N levels and seemed to respond as if N was not limiting growth under low N. Similar results have been recently reported in maize overexpressing *ZmDWF4* grown under field conditions although different N levels were not tested[52]. Uptake rates of N as well as translocation rates did not differ between the OE or WT lines and similar levels of N were found in most tissues measured outside of the grain, suggesting that the OE lines did not demand an increased level of N to support the increased plant growth. The

altered N sensing by higher *DWF4* expression was supported by the expression of N regulated genes which showed a muted response to the removal of N in *TaDWF4-B* OE lines. This expression data suggests that *TaDWF4* overexpression can mitigate nutrient limitations and that other signals such as C or soluble sugars are more important to drive yield gains while lowering the need for N input. This difference in growth was not due to a difference in the photosynthetic capacity, which has been previously suggested as the reason for increased yields[30]. Overall we observed assimilation rates and levels of chlorophyll were similar at high N levels (Fig. 4). The maintenance of C assimilation under limited N conditions may contribute to the increased biomass and yield. The increase in levels of soluble sugars observed in WT plants including glucose possibly suggests altered sensing of required nutrient signals to activate growth and development. This is supported by the higher expression of *TaTOR* under both N levels tested suggesting that *TaDWF4* OE lines maintain growth, where WT would slow down. Further work is required to understand the exact regulation of soluble sugars and other nutrient signals which mediates the increase in growth in *DWF4* overexpressing lines. As BR levels were not directly measured as part of this work it is difficult to determine whether BR levels increased, thereby allowing growth to continue due to altered levels of soluble sugars, or whether the lack of BR pathway inhibition allowed the photosynthetic pathways to produce additional sugars and fuel further growth. There may indeed be other mechanisms outside of the known BR pathway to explain how increased *DWF4* expression allows for growth on otherwise limiting N levels.

Data collected here suggest that higher *TaDWF4* expression in wheat can redirect normal feedback inhibition of N limitation supporting continued plant growth. It would be interesting to determine if similar effects are observed with other common

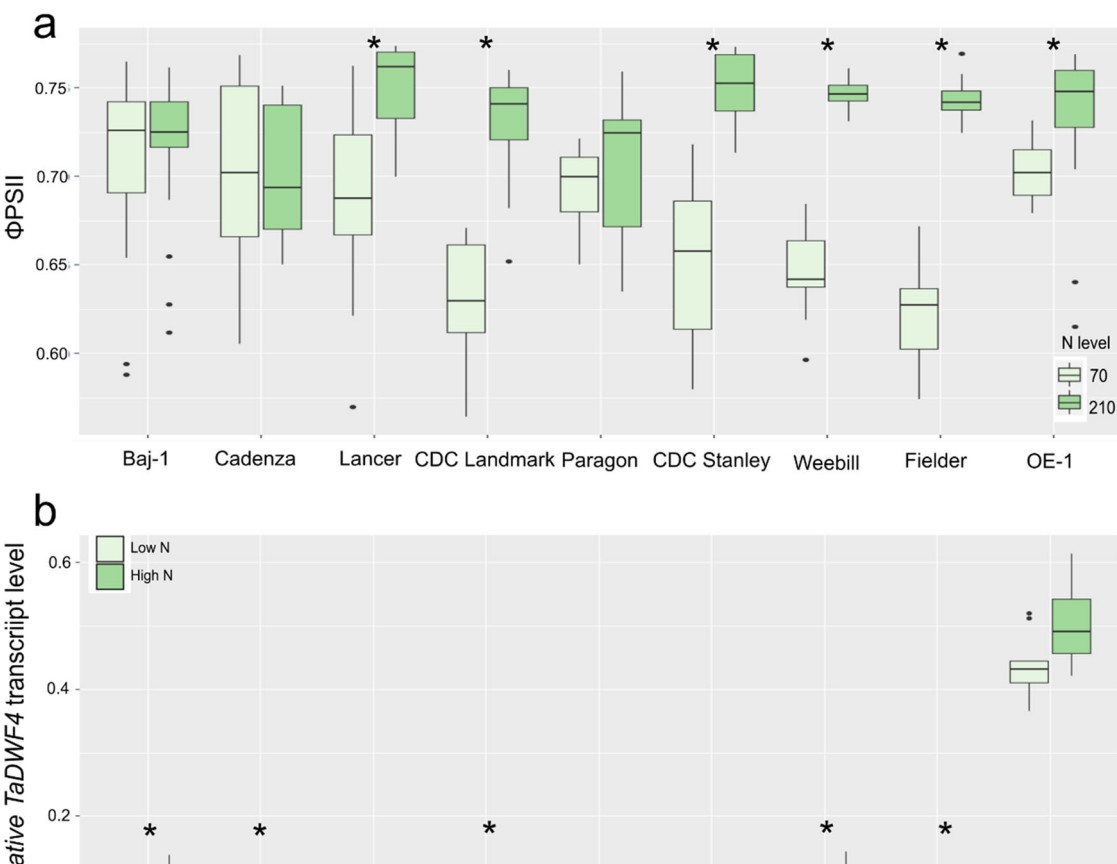

**Fig. 7 No evidence for indirect selection for BR indirectly through breeding. a** Operational PSII efficiency in light-adapted plants grown on two different N levels. Plants were grown on either 70 or 210 kg/ha N equivalent and the fourth leaf was measured for PSII activity. **b** Expression of *TaDWF4* in the shoots of 10-day old wheat plants grown under low N (LN = 0 N) or high N (HN = Full N), the expression shown is relative to *TaUbi* in each cultivar. Data in panels **a** and **b** are shown as the mean values (central line), lower and upper quartiles (box), minimum and maximum values (whiskers), and outliers as individual points. Statistical analysis was performed with ANOVA and post hoc Tukey test, asterisk indicates a significant difference between low and high N levels ($p < 0.05$).

nutrient limitations such as phosphate or potassium. Certainly, a further understanding of how increased *DWF4* expression allows for increased growth and yields will be paramount for translation to other crops and non-GM methods for increasing *DWF4* expression in field-grown crops.

Finally, it has been suggested that breeding for increased BR levels could improve overall yields and improve the tolerance of crop species to various abiotic and biotic stress[28,30,35,41]. Observations here also suggest that wheat breeders are not selecting for increased *DWF4* expression as *TaDWF4* expression was not significantly higher in the more modern spring wheat varieties tested (Fig. 7). There were significant differences in *DWF4* expression between the varieties tested as well as in response to N levels suggesting that variation may exist allowing for selection for higher expression in wheat shoots.

Overall, we have shown that OE of *TaDWF4* in wheat leads to plants that grow as if they need less N, increasing overall NUE. The plants do not sense N as a limitation to growth and keep growing under lower N levels. Based on these results we propose that a doubling of *TaDWF4* expression could allow farmers to reduce the application of N fertilizer by up to 70 kg/ha whilst still producing the same amount of yield per plant as currently grown at standard 210 kg/ha N input level. When this 70 kg/ha saving is

multiplied by the estimated land in which wheat is grown in the UK (1.69 million ha in 2019) there is the potential to reduce $CO_2$ released into the atmosphere from fertilizer production by over 100,000 t[5,53,54]. This is equivalent to the energy needed to power and heat more than 15,000 UK homes for one year[55]. Here we show that it is possible to increase levels of *DWF4* in wheat through genetic modification but that there is also potential to exploit native variation via traditional breeding. This could potentially have a large impact on wheat crop management, reducing fertilizer demands whilst maintaining levels of productivity.

## Materials and methods

**Gene identification.** The *OsDWF4* coding sequence from NCBI (XP_015633105.1) was used as a query for BLAST searches of the wheat genome (IWGSC 2014) and expressed sequence tag (EST) databases in GenBank (http://www.ncbi.nlm.nih.gov/), Ensembl (http://plants.ensembl.org/index.html) and Komugi (https://shigen.nig.ac.jp/wheat/komugi/). Gene prediction was carried out using FGENESH (www.softberry.com). Protein domain prediction was found using InterProScan (https://www.ebi.ac.uk/interpro/). Identification of DWF4 like genes was based on sequences in Ensembl (version 42) using BLASTP and using an e value cut off of <1e$^{-50}$.

**Plasmid construction for genetic modification.** *TaDWF4-B* was synthesized from the public sequence available for the wheat cultivar Chinese Spring with attL1 and attL2 sites for direct recombination into a binary gateway vector. All primers

used in this study are listed in Supplementary Data 1. *TaDWF4-B* was then recombined into the binary vector pSc4ActR1R2 using a Gateway LR Clonase II Kit (Thermofisher) to create pMM90. *TaDWF4-B* was expressed *in planta* from the rice *Actin* promoter and transcripts terminated by the *A. tumefaciens* nopaline synthase terminator (tNOS)[42].

pMM90 was verified by restriction digest and sequencing before being electro-transformed into *A. tumefaciens* strain EHA105 Plasmids were re-isolated from Agrobacterium cultures and verified by restriction digest prior to use in wheat transformation experiments[56].

**Plant materials.** Wheat cv. Fielder (USA) was used for genetic transformation experiments. The spring wheat varieties Baj-1 (India), Cadenza (UK), CDC Landmark (Canada), CDC Stanley (Canada), Lancer (Australia), Paragon (UK), and Weebill (Mexico) were obtained from the Germplasm Resource Unit, UK (www.seedstor.ac.uk/search-browseaccessions.php?idCollection=35), and used in photosynthetic measurements.

**Plant growth conditions.** Wheat cv. Fielder plants were grown in controlled environment chambers (Conviron) at 20 °C day/15 °C night with a 16 h day photoperiod (approximately 400 µE m$^{-2}$ s$^{-1}$) for the harvest of immature embryos for the transformation experiments.

Transgenic lines and corresponding null segregants were grown on TS5 low fertility soil to control total nitrogen with a starting nitrogen level of 0.1 mg/l (Bourne Amenity, Kent, UK). Ammonium nitrate was then added to reach a final concentration in the pots which contained roughly 750 g of dry soil in each pot. Equivalent to field fertilizer application of 70, 140, or 210 kg/ha N which equates to 23.3, 46.6, or 70 mg/pot. Each pot also received 4.2 mg Ca, 2.7 mg K, 0.62 mg Mg, 0.04 mg P, 0.56 mg S, 0.008 mg B, 0.13 mg Fe, 0.015 mg Mn, 0.0012 mg Cu, 0.0024 Mo, 0.0045 Zn, 0.00 mg Na, and 0.63 mg Cl per pot. Plants were grown in a climate-controlled glasshouse with supplemental light for 16 h day and 20 °C/15 °C day–night temperatures.

For transcript abundance measurements and sugar content, wheat seedlings were grown for ten days in 2.2 L pots containing Magnavaca solution containing 3 µM KH$_2$PO$_4$, 3.52 mM Ca(NO$_3$)$_2$, 0.58 mM, KCl, 0.58 mM K$_2$SO$_4$, 0.56 mM KNO$_3$, 0.86 mM Mg(NO$_3$)$_2$ 0.13 mM H$_3$BO$_3$, 5 µM MnCl$_2$, 0.4 µM Na$_2$MoO$_4$, 10 µM ZnSO$_4$, 0.3 µM CuSO$_4$, Fe(NO$_3$)$_3$ and 2 mM MES (pH 5.5), and supplemented with either 0, 0.4 mM, 0.8 mM, or 1.3 mM NH$_4$NO$_3$. Plants were harvested on the tenth day separating tissue in to roots and shoots for analysis.

Plants were grown on Magnavaca solution containing either 0 or 1.3 mM NH$_4$NO$_3$ for ten days in controlled environment chambers (Conviron) at 20 °C day/15 °C night with a 16 h day photoperiod (approximately 400 µE m$^{-2}$ s$^{-1}$).

**Wheat transformation.** Immature seeds were collected 14–20 days post-anthesis (dpa) and immature wheat embryos were isolated and co-cultivated with *A. tumefaciens* for 2 days in the dark[57]. Subsequent removal of the embryonic axis and tissue culture was performed as previously described[58]. Individual plantlets were hardened off the following transfer to Jiffy-7 pellets (LBS Horticulture), potted up into 9 cm plant pots containing M2 compost plus 5 g/l slow-release fertilizer (Osmocote Exact 15:9:9) and grown to maturity and seed harvest in controlled environment chambers, as above.

**DNA analysis of transformed wheat plants.** Plantlets that regenerated under G418 selection in tissue culture, were tested for the presence of the *nptII* gene using QPCR. The *nptII* copy number was assayed relative to a single copy wheat gene amplicon, GaMyb, normalized to a known single copy wheat line[59]. Primers and Taqman probes were used at a concentration of 10 µM in a 10 µl multiplex reaction using ABsolute Blue qPCR ROX mix (Thermofisher) with the standard run conditions for the ABI 7900 HT. The relative quantification, ΔΔ$^{CT}$, values were calculated to determine *nptII* copy number in the $T_0$ and subsequent generations[60]. Homozygous and null transgenic lines were identified on the basis of *nptII* copy number and segregation analysis. WT Fielder plants were null segregates.

**Transcript level analysis.** Total RNA was isolated from both roots and shoots for each nitrogen treatment using an RNeasy Kit (Qiagen). Following treatment with DNaseI (Thermofisher), cDNA synthesis was conducted on 500 ng of total RNA using Omniscript RT Kit (Qiagen). The cDNA was diluted 1:2 with water and 0.5 µL was used as a template in each RT-PCR reaction. Transcripts levels were quantified using SYBR Green JumpStartTaq ReadyMix (SIGMA) with the standard run conditions for the ABI 7900 HT. Three technical replicates were performed on each of the three biological replicates. Two reference genes *TaUbiquitin* and *TaEF1α* were used for the normalization using the ΔΔ$^{CT}$. The sequence of primers used in qPCR assays is shown in Supplementary Data 1.

**Whole plant measurements.** Total shoot dry weight, seed weight (yield per plant), seed number, seed size, and tiller number were measured at maturity and following two weeks of drying at 35 °C. Biological replicates each contained 15 plants per line and were grown until seed maturation and grown in two separate experiments.

NUE was calculated as yield per plant divided by the amount of ammonium nitrate added.

**C and N content determination, N uptake.** Wheat tissues (leaf or grains) were dried at 75 °C for 48 h before grinding. Four grains were placed in 2 mL microfuge tubes with 2 × 5 mm diameter stainless steel beads and shaken in a genogrinder until a fine powder was obtained. Dried and ground samples were carefully weighed (0.5 mg) into tin capsules, sealed, and loaded into the auto-sampler. Samples were analyzed for percentage carbon, percentage nitrogen, $^{12}$C/$^{13}$C (δ$^{13}$C) and $^{14}$N/$^{15}$N (δ$^{15}$N) using a Costech Elemental Analyzer attached to a Thermo DELTA V mass spectrometer in continuous flow mode.

To measure N uptake, roots from 2-week-old seedlings were exposed to $^{15}$NH$_4$$^{15}$NO$_3$ for 5 min, then washed in 0.1 mM CaSO$_4$ for 1 min, harvested, and dried at 70 °C for 48 h. N content and isotopic levels were analyzed as described above. The excess $^{15}$N was calculated based on measurements of δ$^{15}$N and tissue N %. First, the absolute isotope ratio ($R$) was calculated for labeled samples and controls, using $R_{standard}$ (the absolute value of the natural abundance of $^{15}$N in atmospheric N$_2$).

$$R_{sample\ or\ control} = [(\delta^{15}N/1000) + 1] \times R_{standard} \qquad (1)$$

Then, molar fractional abundance ($F$) and mass-based fractional abundance (MF) were calculated

$$F = R_{sample\ or\ control}/(R_{sample\ or\ control} + 1) \qquad (2)$$

$$MF = (F \times 15) \times /[(F \times 15) + ((1 - F) \times 14)] \qquad (3)$$

$$\Delta MF = MF_{sample} - MF_{control} \qquad (4)$$

The excess $^{15}$N in mg in a total tissue was calculated as in

$$Excess\ ^{15}N(g) = \Delta MF \times Tissue\ dw(g) \times Tissue\ N\%/100 \qquad (5)$$

**Chlorophyll measurements.** Leaf chlorophyll content was determined using the method developed by Hiscox et al.[61]. Chlorophyll was extracted from 100 mg of fresh leaf tissue from six independent plants in 10 mL DMSO, mixed for 30 min, and then placed at 4 °C overnight. Extracts were diluted 1:2 with DMSO before absorbance measurements at 645 and 663 nm (spectrophotometer Jenway model 7315, Staffordshire, UK).

**Leaf gas exchange.** Leaf level photosynthesis and stomatal conductance were measured on the youngest fully expanded leaf from the main tiller of at least six wheat plants at the tillering stage (GS23) grown in a controlled environment chamber with 16 h day and 20 °C/15 °C day–night temperatures. Photosynthesis and stomatal conductance were measured using a portable infrared gas analyzer (Licor 6400XP, Licor Environmental). Gas exchange was measured at a PPFD of 1500 µmol m$^{-2}$ s$^{-1}$ (using the LI-COR 6400 LED light source), block temperature was maintained at 22 °C, and humidity was maintained close to 60%. Each leaf was clamped in the Licor chamber, and let to acclimate until CO$_2$ concentrations in the chamber reached a steady-state for 5 min. Gas exchange measurements were recorded under a range of CO$_2$ concentrations (400, 300, 200, 100, 400, 600, 800, 1000, 1200, and 400 ppm).

**Chlorophyll fluorescence measurements and A/C$_i$ curve analysis.** An LI-6800 portable photosynthesis infrared gas analyzer system (LI-COR) equipped with a multiphase flash fluorimeter was used to assess physiological differences for photosynthetic parameters between transgenic and WT wheat plants. Measurements were taken on the fourth leaf of plants grown on TS5 low fertility soil to control total nitrogen with a starting nitrogen level of 0.1 mg/l (Bourne Amenity, Kent, UK). Ammonium nitrate was then added to reach a final concentration in the pots equivalent to field fertilizer application of 70 or 210 kg/ha N. Plants were grown in a climate-controlled chamber with supplemented light (250 µmol m$^{-2}$ s$^{-1}$) for a 16 h day and 20 °C/15 °C day–night temperatures. For chlorophyll fluorescence measurements (Fm'–Fo')/Fm') leaves from at least six plants were pulsed four times to acclimatize tissues and a steady-state reading was taken. For measurements of the A/C$_i$ curve was measured on six plants for each treatment. Ca reference values were 400, 400, 300, 200, 100, 50, 25, 400, 400, 400, 600, 800, 1000, 1200, and 400 µL L$^{-1}$, with a saturating rectangular pulse of 12,000 µmol m$^{-2}$ s$^{-1}$ at each reference point. All leaves were also normalized for the amount of area of the measuring disk. Measurements were carried out on consecutive days between 1 and 8 h post-dawn, measuring three plants total selected at random from each treatment per day.

**Sugar measurements.** Freeze-dried samples were ground to a fine powder with a pestle and mortar. A 50 mg sub-sample of leaf tissue and 25 mg of root tissue was extracted in 1 and 2 ml of ultra-pure water, (Elgastat UHQPS), respectively. The extracts were mixed for 2 min on a maximatic (Jencons) and incubated for 60 min in a water bath at 60 °C and mixed again. The solutions were centrifuged (Sigma

4–16 KS) at 4500 g for 20 min. Five hundred microlitres of the supernatant were pipetted into a Thomson 0.45 µm PTFE filter vial (Thames Restek).

Five microliters of supernatant were injected into a Waters Alliance 2695 HPLC. Sugars were separated on an Ultra amino 100 Å 5 µm 250 × 4.6 mm column (Thames Restek, UK) and detected with a Waters 2414 refractive index detector. The mobile phase was [80:20] [acetonitrile: water] with a flow rate of 1 ml min$^{-1}$, the column was heated to 35 °C. Standards of known amounts of the sugars were injected into the HPLC and Empower$^{TM}$ 3 software was used to produce linear calibration curves in the range of 0.625–10 µg for fructose, glucose, all curves had $r^2$ greater than 0.998. These calibration curves were used to determine the concentration of the sugars found in the samples. Sugar standards were analar grade (Sigma Aldrich UK), solvents were HPLC grade (Fisher Scientific).

**Statistics and reproducibility**. Normal distribution of the data and equality of variance was verified using Shapiro and Levene tests (Lawstat package[62]), respectively. Analysis of variance (ANOVAs) or Wilcox Tests was run using the aov and TukeyHSD functions or Wilcox.test function in the R environment with the null hypothesis of no difference between lines[63]. Tukey's post hoc test was added to identify each significant interaction between the lines tested. Data were plotted using R ggplot2[64].

**Phylogenic trees**. Trees were made using MEGA X[65]. Amino acid sequences were obtained from Ensembl. Amino acid sequences were aligned using the MUSCLE algorithm as part of MEGA X and the tree was constructed using the maximum likelihood method, Jones–Taylor–Thornton model, with 500 bootstrap replications.

**Reporting summary**. Further information on research design is available in the Nature Research Reporting Summary linked to this article.

## Data availability
Seed is available upon request from the corresponding author. Seed materials will be transferred under MTA. Data used to make figures and supplemental figures are deposited in figshare, 10.6084/m9.figshare.19078058[66].

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

## Acknowledgements

This study was supported by BBSRC and the Newton–Bhabha Fund, under the Virtual Joint Centres in Agricultural Nitrogen Use initiative (BB/N013441/1: Cambridge—India Network in Translational Nitrogen (CINTRIN). We thank Dr. Gregory Reeves and Prof. Julian Hibberd (University of Cambridge) for assistance with photosynthetic measurements and CINTRIN members for additional discussions.

## Author contributions

M.C. and S.B. generated the transgenic wheat plants. M.M. performed molecular and biochemical analysis, plus growth, and gas-exchange experiments. S.M.S. performed [15]N uptake experiments. M.M. and S.M.S. performed data analysis on their contributions. M.M. designed the experiments and wrote the paper with input and contributions from S.M.S., H.G., A.R.B., and E.J.W. All authors read and approved the final paper.

## Competing interests

The authors declare no competing interests.
