## [Transparent Peer Review File · Communications Biology]

Reviewers' comments:

Reviewer #1 (Remarks to the Author):

This study generated wheat plants overexpressing BR biosynthesis gene DWF4 and found that the transgenic plants showed increased yield, especially under N limiting conditions. The phenotype was associated with increased photosynthesis and could be attributable to modified carbon partition. Generally, the study is well designed and has great significance as regard to reducing the environmental costs of agricultural production in the context of global climate change. Here are some suggestions for improvement of the manuscript.

- 1) Line 67, nitrogen use efficiency is not an abiotic stress
- 2) Line 79, the sentence is not very clear. Does the author mean that a transcription factor activates the transcription of BZR1 and TOR encoding genes? If so, please give the references.
- 3) Line 87, 'Sugar promotes growth and even if a nutrient becomes limiting, the growth promoting effects of BR is halted.' Please clarify what does the author mean. Does the author propose a hypothesis or shows the facts (if so, give the reference)?
- 4) Line 245, F_v'/F_m' is not the operation efficiency of PSII. Does the author show the data of Phi PSII? The formular for the operation efficiency of PSII should be $(F_m' - F_s)/F_m'$. Please show the details of the protocols of Chl fluorescence measurement.
- 5) Line 259, the sucrose accumulation is undetected in WT but was significantly increased in DWF4-OE plants under N limiting condition. Please explains this in the discussion.
- 6) The legend for fig. 6 is not appropriate. Sugar contents are increased in DWF4-OE plants only in N limiting condition.
- 7) Fig. 6d, for TOR expression in shoots under high N condition, the columns for WT and DWF4-OE plants are labelled with different letters, indicating significant difference. However, in the main text line 267, the author claims that 'no significant difference could be seen in TOR expression under replete conditions'.
- 8) Line 324, the sentence is not quite clear.
- 9) Line 336, improve tolerance to abiotic and biotic stress?
- 10) Line 338, increased DWF4 expression was not significantly higher in more modern spring wheat?

Reviewer #2 (Remarks to the Author):

The current manuscript entitled "Over-expression of the brassinosteroid gene TaDWF4 increases wheat productivity under low and sufficient nitrogen through enhanced carbon assimilation" by Milner et al. reports the positive effects of TaDWF4 overexpression (a brassinosteroid biosynthetic gene) on wheat productivity under nitrogen deficit conditions. The data argued with the classical concept of BR-mediated yield improvement via enhanced photosynthesis, and tried to establish an alternative pathway via the maintenance of photosystem II operating efficiency and modified carbon partitioning. I have read the manuscript with interest. Although the study appears to be comprehensive, many important data are missing. The reader often feels that the authors purposely directed the story-line according to their intention, without considering many other relevant aspects that could be critical for unraveling mechanism, data interpretation and relevant narratives. The authors are requested to address these issues.

After reading the abstract, a reader might have a general understanding that the study is exclusively dedicated to the exploration of the mechanism underlying TaDWF4 overexpression-mediated enhanced tolerance to N starvation. However, the introduction of the manuscript might give a totally different assumption, and it is somewhat confusing to understand the main hypothesis and objectives. This part was mostly dedicated to the evolutionary aspects of wheat and thus the same in the result section. There is a huge gap in coherence between the Abstract and the introduction/results. How the authors distinguish "carbon assimilation" (line 20) and "photosynthesis capacity" (line 21), or else where in the manuscript, is also very confusing and misleading. Therefore, the authors should

improve their narratives with more solid evidence if they would like to establish their argument against the classical concept of BR-mediated CO₂ assimilation and subsequent yield improvement. Please note that operational PSII efficiency (F_v'/F_m') is one of the parameters among many other chlorophyll fluorescence parameters that could give more details about the status of photosystem II under N starvation.

Selection and analysis of the target of rapamycin (TOR) expression is another sign of preconceived notion. In lines, 75-85, the authors, in fact, stated the interaction of TOR and BZR1; however, the most important regulator of BR pathway, BZR1, was not checked either at the transcript or protein level. BZR1 plays a critical role in N starvation response (DOI: 10.1104/PP.18.01028)

In the last paragraph of the introduction, the authors too briefly introduced the role of sugars and tried to connect with BR pathway; however, it is not convincing and the introduction misses the logical basis and hypothesis of the study.

One most important issue is that the author frequently mentioned 'BR levels' (line 293, 303, 329, 331) and ascribed increased BR levels to TaDWF4 overexpression plants; however, they did not analyze the concentrations of BRs. In my opinion, the authors must analyze the concentrations of different BRs, at least the bioactive BRs. Only transcript levels of TaDWF4 are not sufficient to interpret BR levels without biochemical analysis.

Many studies have shown that BR-mediated alteration in leaf angle is an important trait that largely contributes to the photosynthetic capacity and yield of plants. The authors should consider this agronomic trait to understand better the mechanisms of BR-mediated improvement in wheat productivity under N starvation.

Fig. 2: What is the basis for selecting NRT1, NRT2.1, NPF7.1, GDH2? In addition to the transcript levels of these genes, nitrogen metabolism-related genes such as NR, NiR1, NiR2 and GOGAT expression as well as activity of NR and GOGAT should be checked.

Readers will also expect to have an understanding of how TaDWF4 overexpression affected tissue concentrations of other essential macro and micro elements under control and N deficit conditions.

Fig. 4C. The chlorophyll content in wheat leaves is above 15 mg/g FW, is it not too high? Please support your data with reliable references in wheat plants.

The statement (line 64-65) can be supported by a couple of recent references (doi: 10.1007/s00344-020-10098-0; doi: 10.1105/tpc.19.00335)

The authors should also consider the forms of nitrogens (NO₃⁻/NH₄⁺) that affect plant productivity and essentially the wheat crop, and discuss why they chose ammonium nitrate in the current experiment.

Reviewer #3 (Remarks to the Author):

Milner et al. describe in their manuscript that wheat plants overexpressing TaDWF4-B (OE lines) show increased growth performance and grain yield, particularly under limited nitrogen availability. Interestingly, overexpression plants show an attenuated response of genes regulated by N-deficiency. However, grains of OE lines show in general a lower N content, which may negatively impact on the quality.

The study is interesting and well performed. Most data look convincing. However, there are some points that should be addressed:

- 1) The authors describe: "Ammonium nitrate was then added to reach a final concentration in the pots equivalent to field fertiliser application of 70, 140 or 210 kg N /ha." (Line 386) However, please indicate which amount (kg) of soil was used and how much ammonium nitrate (g) was added to each pot. In addition, were also other fertilizers added? If so please describe in detail.
- 2) The authors must provide data how much ammonium and nitrate was still present in the soil after the plants were harvested.

3) The type of BR marker gene used is not the best (Suppl. Fig 3). I agree that EXO has been described to be up-regulated in *A. thaliana* after BL treatment but this is not a commonly used marker gene (even in *A. thaliana*). I would be more convinced if more frequently used marker genes are shown, for instance CPD, ROT3, BR6OX1 and particularly BR6OX2. Particularly the sentence "This would suggest that the over-expression of TaDWF4 leads to higher BR synthesis in wheat shoot" remains totally speculative if no transcript data are provided. Alternative to showing transcript data of the aforementioned genes, the authors could also include quantification of BR levels to support their conclusion.

4) The result for sucrose on wt plants under low N (Fig. 6c) is hard to believe. Are the authors really sure that it was 0 (with a deviation bar)? The authors should measure that again and give references from the literature that wheat leaves do not contain sucrose under N starvation.

5) I cannot believe the chlorophyll contents. The authors show contents in the range of 12-15 mg/g Fw, which corresponds to 12000-15000 mg/kg Fw. However, the highest reported levels are in the range of 1000-2000 mg/kg (see Giuliani et al., 2016, Colors: Properties and Determination of Natural Pigments in Encyclopedia of Food and Health). Moreover, 12-15 mg/g Fw would correspond to approx. 12-15 mg/100 mg DW and thus approx. 12-15%! Such a high value is not plausible.

6) Spacing should be between numbers and units. However, there are dozens of examples where the spacing is missing. Please correct.

Reviewers' comments:

Reviewer #1 (Remarks to the Author):

This study generated wheat plants overexpressing BR biosynthesis gene DWF4 and found that the transgenic plants showed increased yield, especially under N limiting conditions. The phenotype was associated with increased photosynthesis and could be attributable to modified carbon partition. Generally, the study is well designed and has great significance as regard to reducing the environmental costs of agricultural production in the context of global climate change. Here are some suggestions for improvement of the manuscript.

- 1) Line 67, nitrogen use efficiency is not an abiotic stress: we have removed the term abiotic stress
- 2) Line 79, the sentence is not very clear. Does the author mean that a transcription factor activates the transcription of BZR1 and TOR encoding genes? If so, please give the references. We have reworded "The BR and TOR pathways directly interact though the activation of the transcription factor which activates BR related genes, BZR1, and the TOR kinase." to read "The BR and TOR pathways directly interact though the BR related transcription factor, BZR1, and the TOR kinase."
- 3) Line 87, 'Sugar promotes growth and even if a nutrient becomes limiting, the growth promoting effects of BR is halted.' Please clarify what does the author mean. Does the author propose a hypothesis or shows the facts (if so, give the reference)?

We have removed the sentence for clarity.

- 4) Line 245, F_v'/F_m' is not the operation efficiency of PSII. Does the author show the data of Phi PSII? The formular for the operation efficiency of PSII should be $(F_m' - F_s)/F_m'$. Please show the details of the protocols of Chl fluorescence measurement. We measured the ϕ_{PSII} which was calculated as $Y(II)$ which $= (F_m' - F_o') / F_m'$ for light adapted leaves. We have also added to the methods section under Chlorophyll fluorescence measurements to properly disclose our methods. We apologize for this oversight. We have also added to line 251 of the revised m/s to clarify the measurements taken.

- 5) Line 259, the sucrose accumulation is undetected in WT but was significantly increased in DWF4-OE plants under N limiting condition. Please explains this in the discussion.

We have decided to take out the data on sucrose but added root and shoot data for glucose and fructose this time measured by HPLC to give a clear measurement in figure 6. We decided to remove the sucrose data as new experiments from the reviewers suggestions and follow on work has indicated the difference in levels observed might be more involved in the delayed development of WT plants under low N conditions and not due to DWF4 specifically.

6) The legend for fig. 6 is not appropriate. Sugar contents are increased in DWF4-OE plants only in N limiting condition.

We have changed the legend to “Figure 6: Root and shoot glucose and fructose content in OE-1 and WT wheat plants grown under high N or low N conditions in hydroponic solution, leads to increased transcript levels of TaTOR.”

7) Fig. 6e, for TOR expression in shoots under high N condition, the columns for WT and DWF4-OE plants are labelled with different letters, indicating significant difference. However, in the main text line 267, the author claims that ‘no significant difference could be seen in TOR expression under replete conditions.

This has been corrected on line 271 to read as significant differences were seen under both nitrogen treatments.

8) Line 324, the sentence is not quite clear.

We have changed the text to try and clearly state how sugar levels may play a role in the interaction of BR and TOR pathways.

9) Line 336, improve tolerance to abiotic and biotic stress?

We have changed the wording slightly to try and clarify the point being made. As in the literature modification of BR levels has been shown to increase drought and salt tolerance as well as decrease susceptibility to various pathogens as referenced in the discussion on line 342.

10) Line 338, increased DWF4 expression was not significantly higher in more modern spring wheat? We have tried to clarify the statement as no clear trend was seen for higher DWF4 expression in modern spring wheats. As there was a significant difference in the levels of DWF4 seen between the spring wheat panel tested outside of the transgenic line OE-1. There were however significant differences seen between lines under the two N levels as indicated in Figure 7B. As Fielder a variety originally released in the 1970s actually had the highest DWF4 expression under the high N treatment but one of the lowest under low N not clear selection has been happening to keep DWF4 expression high through modern breeding.

Reviewer #2 (Remarks to the Author):

The current manuscript entitled “Over-expression of the brassinosteroid gene TaDWF4 increases wheat productivity under low and sufficient nitrogen through enhanced carbon assimilation” by Milner et al. reports the positive effects of TaDWF4 overexpression (a brassinosteroid biosynthetic gene) on wheat productivity under nitrogen deficit conditions. The data argued with the classical concept of BR-mediated yield improvement via enhanced photosynthesis, and tried to establish an alternative pathway

via the maintenance of photosystem II operating efficiency and modified carbon partitioning. I have read the manuscript with interest. Although the study appears to be comprehensive, many important data are missing. The reader often feels that the authors purposely directed the story-line according to their intention, without considering many other relevant aspects that could be critical for unraveling mechanism, data interpretation and relevant narratives. The authors are requested to address these issues.

After reading the abstract, a reader might have a general understanding that the study is exclusively dedicated to the exploration of the mechanism underlying TaDWF4 overexpression-mediated enhanced tolerance to N starvation. However, the introduction of the manuscript might give a totally different assumption, and it is somewhat confusing to understand the main hypothesis and objectives. This part was mostly dedicated to the evolutionary aspects of wheat and thus the same in the result section. There is a huge gap in coherence between the Abstract and the introduction/results.

How the authors distinguish “carbon assimilation” (line 20) and “photosynthesis capacity” (line 21), or else where in the manuscript, is also very confusing and misleading. Therefore, the authors should improve their narratives with more solid evidence if they would like to establish their argument against the classical concept of BR-mediated CO₂ assimilation and subsequent yield improvement. Please note that operational PSII efficiency (F_v'/F_m') is one of the parameters among many other chlorophyll fluorescence parameters that could give more details about the status of photosystem II under N starvation.

In our context carbon assimilation is the amount of CO₂ being fixed into a C backbone under the given conditions. This is the A or assimilation measurements were talk about in Figures 4 and 5. In the context of figure 5 the growth Photosynthesis capacity is the assimilation under optimum light and higher levels of CO₂ which is not how plants are grown in a chamber or the field. This is the data presented in Figure 4A and B while this is still measuring C assimilation the parameters under which these measurements are taken in artificial thus termed photosynthesis capacity. These conditions include light levels approx. five times higher than the normal light levels used in glasshouses or controlled environment chambers as we vary the amount of CO₂ available to the leaf.

Selection and analysis of the target of rapamycin (TOR) expression is another sign of preconceived notion. In lines, 75-85, the authors, in fact, stated the interaction of TOR and BZR1; however, the most important regulator of BR pathway, BZR1, was not checked either at the transcript or protein level. BZR1 plays a critical role in N starvation response (DOI: 10.1104/PP.18.01028).

We have attempted to clone and characterize the interaction at the protein level however *TaTOR* seems to be very toxic in E coli cells and is not easily cloned. To further address the concern, as I am sure the reviewer is aware there is not a high degree of homology between AtBZR1 and monocot BZR1 type proteins. (DOI: <https://doi.org/10.1073/pnas.0706386104> and <https://doi.org/10.1104/pp.19.00100>). So we have measured TaBZR2 transcripts located on chromosome 2 to try and address the reviewers concerns and added this as a suppl. figure to the manuscript (Suppl. Figure X). We have also mentioned in the text the results of this expression to further support the statements made.

In the last paragraph of the introduction, the authors too briefly introduced the role of sugars and tried to connect with BR pathway; however, it is not convincing and the introduction misses the logical basis and hypothesis of the study.

We have removed the sentence referred to on line 88 and 89 of the previous version of the m/s submission to avoid the confusion.

One most important issue is that the author frequently mentioned 'BR levels' (line 293, 303, 329, 331) and ascribed increased BR levels to TaDWF4 overexpression plants; however, they did not analyze the concentrations of BRs. In my opinion, the authors must analyze the concentrations of different BRs, at least the bioactive BRs. Only transcript levels of TaDWF4 are not sufficient to interpret BR levels without biochemical analysis.

For the past eight months we have attempted to measure BR levels directly in both the OE lines and WT wheat, but we are unable to accurately measure BR levels as only a few labs worldwide have ever been able to measure this compound directly. Thus in an attempt to satisfy the reviewers comment we have added the relative expression of a few more BR related genes to add further support. It should be noted that we originally tried to address this concern by measuring the BR responsive gene EXOI as part of the original submission (Suppl figure 3B). But as another reviewer also raised similar concerns so we have now included expression of more BR related genes as a supplemental figure which included the aforementioned DWF3/CPD/CYP90A, BRI2 and BES1 as an expanded Supplemental Figure 3.

We have also changed the text slightly to tone down the BR levels concern raised on the previous version of the m/s lines 293, 303, 329 and 331.

Many studies have shown that BR-mediated alteration in leaf angle is an important trait that largely contributes to the photosynthetic capacity and yield of plants. The authors should consider this agronomic trait to understand better the mechanisms of BR-mediated improvement in wheat productivity under N starvation.

While we agree with the reviewer we feel this is somewhat outside of the scope and message of the submitted manuscript. We have actually done these experiments using publicly available TILLing lines but due to the location of the mutations and the redundancy of DWF4 type genes in wheat having a clear genotype on the progeny difficult. Thus we chose not to include this data in the current manuscript for this reason. We are happy to share this with the reviewer at a later date as we have analyzed the TILLing lines for the three homeologues and the B homeologue clearly shows the erect leaf phenotype ~25 percent of the time in two independent premature stop TILLing lines.

Fig. 2: What is the basis for selecting NRT1, NRT2.1, NPF7.1, GDH2? In addition to the transcript levels of these genes, nitrogen metabolism-related genes such as NR, NiR1, NiR2 and GOGAT expression as well as activity of NR and GOGAT should be checked. As stated in the text of the m/s NRT1 and NRT2.1 are involved in the low and high affinity uptake of NO₃ from the soil, NPF7.1 is involved in N translocation through the plant and GDH2 is involved in recycling of carbon backbones to keep photosynthesis going. That is why the genes were selected. As the reviewer correctly points out there are a number of other

genes which could be studied for their response to altered DWF4 expression, but we feel this will not greatly add to the current manuscript.

Readers will also expect to have an understanding of how TaDWF4 overexpression affected tissue concentrations of other essential macro and micro elements under control and N deficit conditions.

We would like to thank the reviewer for the comment but as we did not see a difference in the N concentration of the leaf material or differences in uptake or translocation of N we would question why the reviewer would want to see modification of other elements. The only place a difference is seen in N levels is in the grain. As this is now part of a subsequent paper understanding why this difference in N grain levels exists. Thus we have decided not to include this data as we do not feel this would greatly add to the message of the current manuscript.

Fig. 4C. The chlorophyll content in wheat leaves is above 15 mg/g FW, is it not too high? Please support your data with reliable references in wheat plants.

The reviewer is correct we are an order of magnitude too high. This has been corrected in figure 4C.

The statement (line 64-65) can be supported by a couple of recent references (doi: 10.1007/s00344-020-10098-0; doi: 10.1105/tpc.19.00335)

These references have now been included to address the reviewers concern.

The authors should also consider the forms of nitrogens ($\text{NO}_3^-/\text{NH}_4^+$) that affect plant productivity and essentially the wheat crop, and discuss why they chose ammonium nitrate in the current experiment.

Again the reviewer makes a good point but this is already a fairly long paper and we would argue this is part of future work and outside of the points trying to be made as part of the submitted manuscript.

Reviewer #3 (Remarks to the Author):

Milner et al. describe in their manuscript that wheat plants overexpressing TaDWF4-B (OE lines) show increased growth performance and grain yield, particularly under limited nitrogen availability. Interestingly, overexpression plants show an attenuated response of genes regulated by N-deficiency. However, grains of OE lines show in general a lower N content, which may negatively impact on the quality.

The study is interesting and well performed. Most data look convincing. However, there are some points that should be addressed:

1) The authors describe: "Ammonium nitrate was then added to reach a final concentration in the pots equivalent to field fertiliser application of 70, 140 or 210 kg N /ha." (Line 386) However, please indicate which amount (kg) of soil was used and how much ammonium nitrate (g) was added to each pot. In addition, were also other fertilizers added? If so please describe in detail.

We have added the information requested unfortunately we did not weight the pots but can tell you that 1 L pots filled with the low fertility soil weigh approx. 750 g dry or about 10 cm² volume. We have added the weight of the average pot along with the amount of fertilizer (23.3 mg per pot for each 70 kg/ha equivalent) added per pot to the methods as requested.

2) The authors must provide data how much ammonium and nitrate was still present in the soil after the plants were harvested.

Unfortunately this was not measured and would require a full growth experiment to collect this data. We however did measure the N in the leaves and grain to try and understand the amounts of N moving through the plant as well as uptake and translocation rates of N. We feel that while this compromise does not give the reviewer the exact data they would like it should help readers reproduce or improve on the current experiments presented as part of this manuscript.

3) The type of BR marker gene used is not the best (Suppl. Fig 3). I agree that EXO has been described to be up-regulated in *A. thaliana* after BL treatment but this is not a commonly used marker gene (even in *A. thaliana*). I would be more convinced if more frequently used marker genes are shown, for instance CPD, ROT3, BR6OX1 and particularly BR6OX2. Particularly the sentence "This would suggest that the over-expression of TaDWF4 leads to higher BR synthesis in wheat shoot" remains totally speculative if no transcript data are provided. Alternative to showing transcript data of the aforementioned genes, the authors could also include quantification of BR levels to support their conclusion.

We have now added expression of three more BR associated genes DWF3/CPD, BRI2 and BES1 as an expanded Suppl. figure 3 and incorporated this data into the text of the manuscript. We did not choose to measure BR6OX type genes as there appears to be a large increase in copy number in wheat as we identified four genes similar on chromosome 2A all next to each other, and a similar 5 genes each on chromosomes 2B and D all in sequential order of the chromosome. ROT3 was also not included as a clear homologue/orthologue could not be identified in wheat using BLAST.

4) The result for sucrose on wt plants under low N (Fig. 6c) is hard to believe. Are the authors really sure that it was 0 (with a deviation bar)? The authors should measure that again and give references from the literature that wheat leaves do not contain sucrose under N starvation.

There was a small amount of sucrose detected in a few of the samples tested, but the average sucrose level came out to nearly 0 when taking into account the other sugar types also measured by the subtractive method used. But having again measured the sugars as requested by the reviewer this time by HPLC we think there is a developmental aspect to sucrose levels at the time point we have chosen and since WT is slightly developmentally delayed due to the low levels of N for the treatment might be a confounding factor between the difference seen between WT and the OE line for levels of sucrose, although we can clearly see sucrose in mature WT leaves. Thus we have decided to drop to reporting of sucrose levels as part of this work as we are looking in to this in greater detail. That being said there is precedence for this in the literature as we have added to the text to highlight this point. We have included a recent paper looking a sulphur deficiency and similar results where lowering sulphur in the media decreased sucrose levels similar to what we saw in our experiments. (Dong, Y., Silbermann, M., Speiser, A. et al. Sulfur availability regulates plant growth via glucose-TOR signaling. *Nat Commun* 8, 1174 (2017). <https://doi.org/10.1038/s41467-017-01224-w>).

5) I cannot believe the chlorophyll contents. The authors show contents in the range of 12-15 mg/g Fw, which corresponds to 12000-15000 mg/kg Fw. However, the highest reported levels are in the range of 1000-2000 mg/kg (see Giuliani et al., 2016, Colors: Properties and Determination of Natural Pigments in Encyclopedia of Food and Health). Moreover, 12-15 mg/g Fw would correspond to approx. 12-15 mg/100 mg DW and thus approx. 12-15%! Such a high value is not plausible.

The reviewer is correct we were off by an order of magnitude and this has been corrected in figure 4C.

6) Spacing should be between numbers and units. However, there are dozens of examples where the spacing is missing. Please correct.

We have gone through the manuscript and corrected this at the reviewers request.

Reviewers' comments:

Reviewer #1 (Remarks to the Author):

The manuscript has been substantially improved after thorough revision. I have no major concerns, except that the chlorophyll fluorescence parameter used by the author is not operational efficiency of PSII. As the formula is for maximum efficiency of PSII under light. Please revise this mistake.

(Regarding Reviewer #3's comments)

Because the negative feedback of BR synthesis by the BR signaling effector BZR1 or BES1, the control of BR homeostasis is quite complex. It is not straightforward to evaluate the strength of BR signaling simply by determine the expression of BR biosynthesis genes. Meanwhile, even the quantification of endogenous BR content is not sufficient to indicate the BR signaling. I noticed that N deficiency resulted in a downregulation of DWF4 gene, however, expression of DWF4 driven by the promoter of the housekeeping gene actin led to an improvement of growth performance under N deficient stress. In addition, the difference in the photosynthetic capacity as determined by the A/Ci curve and the above ground biomass between WT and OE plants was more significant under N deficiency. These results are interesting and suggest that the inhibition of BR synthesis limits the metabolism and growth of plants under N deficiency. However, before getting a conclusion, I suggest the author to compare the transcript of DWF4 between WT and OE plants under both N deficient and sufficient conditions. If the growth performance and photosynthesis correlated with the DWF4 transcript, I think the author can conclude at least that increasing the expression of DWF4 improves photosynthesis and performance under N deficiency.

Reviewer #2 (Remarks to the Author):

Recommendation: Accept

Reviewer #3 (Remarks to the Author):

The authors have included the missing information. However, the data shown in new Suppl. Fig. 3b are confused: originally, the authors stated that overexpression of TaDWF4 increases BR levels. However, the data provided now do not seem to support this hypothesis. It has been shown multiple times that expression of CPD/DWF3/CYP90A goes down if BR accumulates or under BR-treatment. In *A. thaliana* BZR1 has been shown to be up-regulated by BR-treatment while BES1 did not react (De Rybel et al., 2009, Suppl. Fig. S2). Here opposite results are shown. Since the expression data are ambiguous, I agree to reviewer 2 that BR levels must be measured. I do not think that it is sufficient to tone down that BR levels are upregulated. In fact, this is the mechanism how overexpression of TaDWF4 is believed to impact on plant growth, development, performance etc. and thus solid data must be provided. Otherwise it is difficult to interpret the results. In the legend the authors write TaBZR1 while in the Figure BZR2 is written. I think, they should also be more consistent with using "Ta" and use that also in the Figure.

Firstly we would like to thank the reviewers for their time and input in reviewing and improving our manuscript. We have addressed the concerns raised in the revised manuscript and hope that you will consider it is now acceptable for publication in Communications Biology.

To address reviewer's 1 comments

We have corrected the Y axis and legend in figure 5 as advised by the reviewer to correct our error.

To address reviewer 1's point made regarding reviewer 3's main concern we have modified the discussion to reflect that this is most likely not a simple straight forward response to high DWF4 expression. Reviewer 3 suggested we measure DWF4 expression under high and low N levels in both the OE line and WT plants. We have already provided this data in figure 7B a comparison of DWF4 expression in a diverse panel of wheats, including Fielder and the overexpression line OE-1, grown under high and low N levels. The same conditions in which the carbon assimilation data and maximum efficiency of PSII under high and low N were also measured to validate the expression data presented in figure 7B.

To address reviewers 3 comments on the originally missing information which was subsequently included in suppl fig 3. We feel that there is a fundamental difference to the overexpression of a part of the BR pathway (DWF4) versus that of BL being applied directly to a plant. We would point to data in Arabidopsis which DWF4 is over expressed (Choe et al, 2001 <https://doi.org/10.1046/j.1365-313x.2001.01055.x>). In figure 5 DWF3 which the reviewer states CPD/DWF3/CYP90A is altered by BR levels is not differently expressed in the DWF4 OE line (AOD4). This is similar to what we reported as part of this work. The De Rybel et al., 2009 Suppl figure is measured on 3-day old seedlings treated with BL for 6 hours. This is very different than a stable transformed line overexpressing DWF4 and may explain the discrepancy between expression levels of the genes the reviewer highlights. In wheat, Cui et al 2019 (doi: 10.1104/pp.19.00100) shows that BR added to leaf or root tissue in two-week-old wheat seedlings exhibit transient responses to BR and that by 24 hours, *TaBZR2* expression is back to "control" levels. We used these same primers used by Cui et al., to measure *TaBRZ2* in our overexpression lines. Based on the reviewers comment and this evidence we have further down played the BR levels in the text to try and not mislead readers about BR levels as this was not directly measured but rather modification of *DWF4* expression. We have also contacted two groups with hormone measurement capability regarding direct measurement of BRs in the material, and unfortunately it was not possible to accomplish in a realistic timeframe and cost basis.

Finally we have added the wheat species identifier (Ta) to the gene expression figures as requested.

We hope the remaining reviewers agree that we have addressed the concerns as far as technically possible and that the manuscript is suitable for publication.

Best

Dr. Matthew Milner

REVIEWERS' COMMENTS:

Reviewer #1 (Remarks to the Author):

accept

Reviewer #3 (Remarks to the Author):

The manuscript is clearly interesting since the authors show that overexpression of TaDWF4 in wheat can enhance yield, particularly under N limitation, significantly. However, no clear data for the mechanism behind that are presented. The manuscript suggest that this is achieved by enhancing BR biosynthesis. For instance, the first sentences auf the abstract clearly suggest that: "Brassinosteroids, (BR), are a group of phytohormones ... that have been demonstrated to regulate several agronomic traits." "DWF4 encodes a cytochrome P450 that catalyses a rate-limiting step in BR synthesis." However, the authors fail showing that BR biosynthesis is indeed enhanced in the OE lines. The authors show that there are 7 homologues of At DWF4 in wheat. They picked the one showing highest expression. They also could not show that the OE plants have increased BR levels (as it is expected if really the homologue of DWF4 was overexpressed). The results from the BR marker genes are not clear because some genes showed a response, others did not. They did no show that the selected protein has indeed campesterol hydroxylase activity. Since there are 7 copies it is of course ot necessary that all of the have the function known for AtDWF4, namely campesterol hydroxylase activity.

Since there is no proof that the selected TaDWF4 has really a function in BR biosynthesis the manuscript remains speculative in large parts. I do not doubt the physiological studies. However, the mechanism responsible for these effects is difficult to address with the data shown. In the worst case it could be that the overexpressed gene has another function and the effects are independent from BRs at all. This is clearly a drawback of this manuscript.